# Involvement of Flagellin in Kin Recognition between *Bacillus velezensis* Strains

Yan Liu,[a,b] Rong Huang,[a,b] Yuqi Chen,[a,b] Youzhi Miao,[a,b] Polonca Štefanič,[c] Ines Mandic-Mulec,[c] Ruifu Zhang,[a,b] Qirong Shen,[a,b] Zhihui Xu[a,b]

[a]Jiangsu Provincial Key Lab of Solid Organic Waste Utilization, Jiangsu Collaborative Innovation Center of Solid Organic Wastes, Nanjing Agricultural University, Nanjing, Jiangsu, Peoples R China

[b]Educational Ministry Engineering Center of Resource-Saving Fertilizers, The Key Laboratory of Plant Immunity, Nanjing Agricultural University, Nanjing, Jiangsu, Peoples R China

[c]Department of Microbiology, Biotechnical Faculty, University of Ljubljana, Ljubljana, Slovenia

**ABSTRACT** Kin discrimination in nature is an effective way for bacteria to stabilize population cooperation and maintain progeny benefits. However, so far, the research on kin discrimination for *Bacillus* still has concentrated on "attack and defense" between cells and diffusion-dependent molecular signals of quorum sensing, kin recognition in *Bacillus*, however, has not been reported. To determine whether flagellar is involve in the kin recognition of *Bacillus,* we constructed *Bacillus velezensis* SQR9 assembled with flagellin of its kin and non-kin strains, and performed a swarm boundary assay with SQR9, then analyzed sequence variation of flagellin and other flagellar structural proteins in *B. velezensis* genus. Our results showed that SQR9 assembled with flagellin of non-kin strains was more likely to form a border phenotype with wild-type strain SQR9 in swarm assay than that of kin strains, and that non-kin strains had greater variation in flagellin than kin strains. In *B. velezensis*, these variations in flagellin were prevalent and had evolved significantly faster than other flagellar structural proteins. Therefore, we proposed that flagellin is an effective tool partly involved in the kin recognition of *B. velezensis* strains.

**IMPORTANCE** Kin selection plays an important role in stabilizing population cooperation and maintaining the progeny benefits for bacteria in nature. However, to date, the role of flagellin in kin recognition in *Bacillus* has not been reported. By using rhizospheric *Bacillus velezensis* SQR9, we accomplished flagellin region interchange among its related strains, and show that flagellin acts as a mediator to distinguish kin from non-kin in *B. velezensis*. We demonstrated the polymorphism of flagellin in *B. velezensis* through alignment analysis of flagellin protein sequences. Therefore, it was proposed that flagellin was likely to be an effective tool for mediating kin recognition in *B. velezensis*.

**KEYWORDS** *Bacillus velezensis*, kin recognition, bacterial flagellum, variation of flagellin sequences, flagellin tertiary structure prediction, variation of hag gene

In the natural environment, bacteria have a variety of multicellular cooperation lifestyles including biofilm (1), quorum sensing (2), swarming (3), sliding motility (4), production of extracellular enzymes (5), and labor division (6). It is advantageous for them to increase nutrient acquisition, resist unfavorable environments, avoid predation, and enhance the chance of survival and reproduction (7). A mechanism known as kin discrimination is used by bacteria to promote cooperative behavior among populations (8, 9). Kin discrimination means that bacteria treat organisms differently based on their kinship. They cooperate with organisms recognized as kin while competing with organisms recognized as non-kin (10–12). The 4 genes, *gyrA*, *rpoB*, *recA*, and *dnaJ*, have been used to determine the kinship of 39 *Bacillus subtilis* strains. In an analysis of the 4

Address correspondence to Zhihui Xu, xzh2068@njau.edu.cn.

The authors declare no conflict of interest.

genes, it was found that bacteria with similarities over 99.5% could be characterized as kin. The 2 kin strains merged on the swarm plate, while non-kin strains formed a boundary. In addition, kin strains can form a mixed biofilm on the root surface, but non-kin are not (13). Afterward, it was also discovered that the strains that do not produce surfactin could exploit the surfactin produced by the kin strains to restore part of the swarming motility ability, while the non-kin strains cannot provide this help (14). Kin discrimination is greatly important for bacteria to survive and development and fitness capabilities, but concrete research of molecular mechanisms is limited.

The research on bacterial kin discrimination mechanism is mostly focused on Gram-negative bacteria, such as *Proteus mirabilis* (15), *Escherichia coli* (16), *Myxococcus xanthus* (17, 18), and *Pseudomonas aeruginosa* (19); there are few studies on kin discrimination of Gram-positive bacteria, mainly including *Staphylococcus aureus* (20) and *B. subtilis* (21). In his review, Wall roughly summarizes the kin discrimination process of bacteria into 3 steps: receptor-ligand or receptor-receptor binding to recognition, signal or biochemical perception, and behavioral response (11). The research on kin discrimination for *Bacillus* has so far concentrated on "attack and defense" between cells, which is the second step of kinship discrimination. The kin discrimination system is highly complicated, involving genes encoding for proteins with a variety of functions, including microbial attack and defense genes: *wapAI*, *sdpABC*, *sdpIRs* and *skfA-H*, toxin, and immunity genes: *sunA* and *bacA*, and antibiotic attack-related genes: *lytST*, *yvrHB* and *sigW*, etc (10). The first step of kin recognition remains at the level of quorum sensing, which is a cell population behavior mediated by diffusion-dependent molecular signals (22), however, there is no report on whether there is a kin recognition mechanism in *Bacillus*.

Flagella is a complex structure regulated by a series of genes in bacteria (23). It is divided into 3 parts: trans-membrane basal body, hooks, and filaments, up to 15 $\mu$m in length (24). Flagella participates in both swimming and swarming motility, which are 2 important forms of bacteria movement (3, 25). Filaments are hollow tubes formed by the polymerization of flagellin monomers and are located outside the cell (26). Flagellin is divided into 4 domains: D0, D1, D2, and D3. The D0 and D1 domains are involved in the aggregation of flagellin monomers into flagellar filaments, which are highly conserved in various flagellated bacteria and are a microbe-associated molecular pattern. Flagellin D2 and D3 domains are not present in all flagellated bacteria, and their deletion weakens flagellin's intrinsic antigenicity but does not interfere with its immunostimulatory effects (27–29). Research has been conducted on the recognition and induction of various cells to bacterial flagella in recent years. For example, cells in vertebrates have evolved pattern recognition receptors, including TLR5 and NLRC4, which recognize the highly conserved region of flagellin as a danger signal (30). Plant cells recognition receptor FLS2, which can detect the 22 amino acids at the conserved N-terminal of flagellin, triggering a series of immune responses (31). A specific methanogenic archaeon perceived a bacterial flagellum protein and activated its methanogenesis, which suggests that the bacterium communicates with the archaeon by using its flagellum (32). However, the mechanism of bacteria-bacteria communication via flagella is poorly understood.

The filaments that extend outside the bacterial cell are several times longer than the cell body, and they might be used as a medium for mediating communication between bacteria, as the first step in kin discrimination. To date, it has not been reported whether flagellar are involved in bacterial recognition, or whether they are related to the evolution of strains of bacteria. In this study, a set of *hag* genes were exchanged to *Bacillus velezensis* with different phylogenetic distances from *B. velezensis* SQR9, to learn how differences in filament lead to a difference in recognition. In addition, we also performed an in-depth analysis of the sequence variation of flagellin and other flagellar structural proteins within *B. velezensis* species. The results showed that differences in flagellar filaments would alter the recognition phenotype of the strain on semi-solid plates, suggesting flagellin is involved in kin recognition of *B. velezensis* strains.

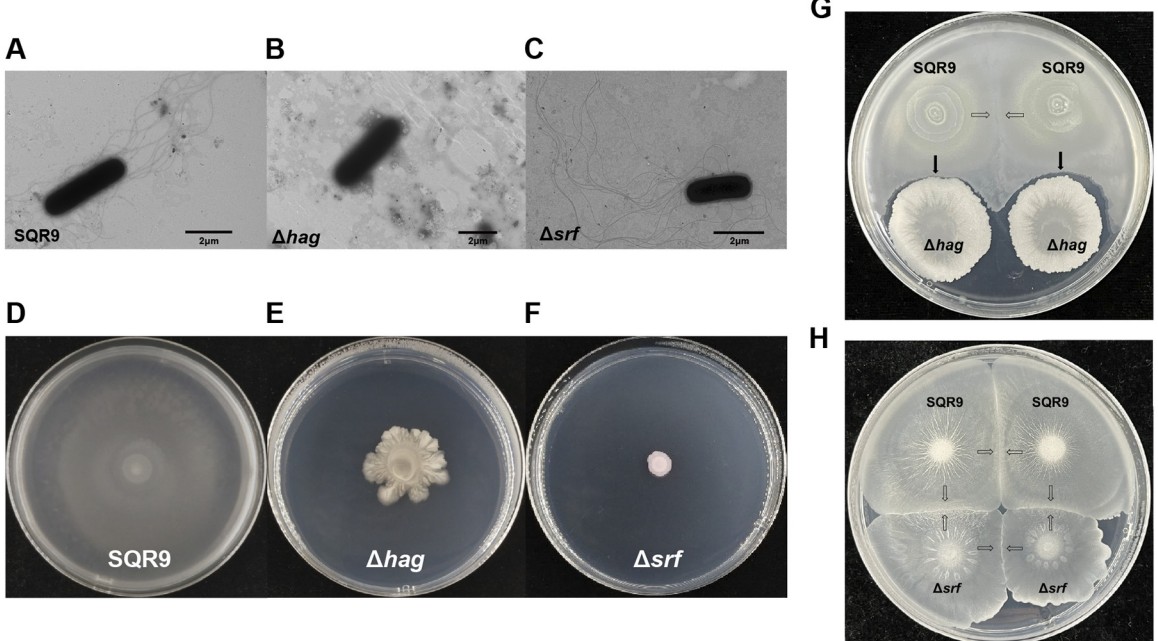

**FIG 1** Deletion of *hag* gene in *B. velezensis* SQR9 resulted in the loss of flagella filaments and phenotype changes in the swarm boundary assay with itself. The transmission electron microscopy photos of the SQR9wt (A) and Δ*hag* mutant (B) showed that deletion of the flagellin *hag* gene prevented the strain from growing flagellar filaments, and deletion of the *srf* gene did not affect bacterial flagella assembly (C). The HT7700 transmission electron microscope (TEM) was operated at 80 kV to observe and photograph. The swarm phenotype of the SQR9wt (D), Δ*hag* (E), and Δ*srf* (F) displayed that the absence of flagellar filaments or surfactin renders the strain incapable of the swarm. (G) the loss of flagellar filaments hindered kin discrimination between SQR9wt and Δ*hag* in swarm assay. (H) Synthesis defect of surfactin did not affect kin discrimination between SQR9wt and Δ*srf* in swarm assay. The white arrows represented merge phenotype, while the black arrows represented boundary phenotype. All results are representative of three experiments.

## RESULTS

**The lack of flagellin impairs kin recognition of *B. velezensis* SQR9.** The flagella are very important motility organs for the bacterium, and the bacteria without flagella are not able to swim and swarm (3, 25). *B. velezensis* SQR9 is a plant-growth promoting rhizobacteria (PGPR) strain, isolated from cucumber rhizosphere soil. *B. velezensis* SQR9 mutant strain Δ*hag* lacked flagellar filaments and did not swarm on a Semi-solid medium (Fig. 1A, B, D, and E). The strain SQR9wt (wild type) merged with itself on the swarming plate (Fig. 1G, white arrows), while cannot merge with mutant Δ*hag* without flagellar filaments, and even it avoided the surrounding area of Δ*hag* to grow (Fig. 1G, black arrows). Moreover, we used the Δ*srf* (the surfactin synthetic gene mutant) as a control with motility mutants different than Δ*hag*. Results showed that Δ*srf* mutant of *B. velezensis* SQR9 has the complete and normal flagella as the wild-type strain (Fig. 1C), but loses its swarming ability (Fig. 1F). When the Δ*srf* mutant meets the SQR9 wild-type during swarm assay, they merged on the swarm plate (Fig. 1H). These results indicated that flagellin is potentially involved in kin recognition for *B. velezensis* SQR9. Additionally, we observed that the Δ*srf* mutant with impaired swarming ability could exploit surfactin from wild-type as biosurfactant to restore its swarming ability (Fig. 1H), and these observations were consistent with the formerly report by Nicholas A. Lyons and Roberto kolter (33). Based on the results, kin recognition and cooperation were hindered between the mutant strain and the wild-type strain, suggesting that bacterial flagellar filaments might play a role in kin recognition in *B. velezensis*.

**B. velezensis SQR9 assembled with "non-kin" flagella now behaved as non-kin when confronted with the wild-type in swarming assay.** To study how flagellar filaments affect kin recognition ability in *B. velezensis*, we collected 20 strains belonging to *B. velezensis*, constructed a phylogenetic tree, and performed a swarm boundary assay between SQR9 and other 19 strains, to characterize kinship distance between

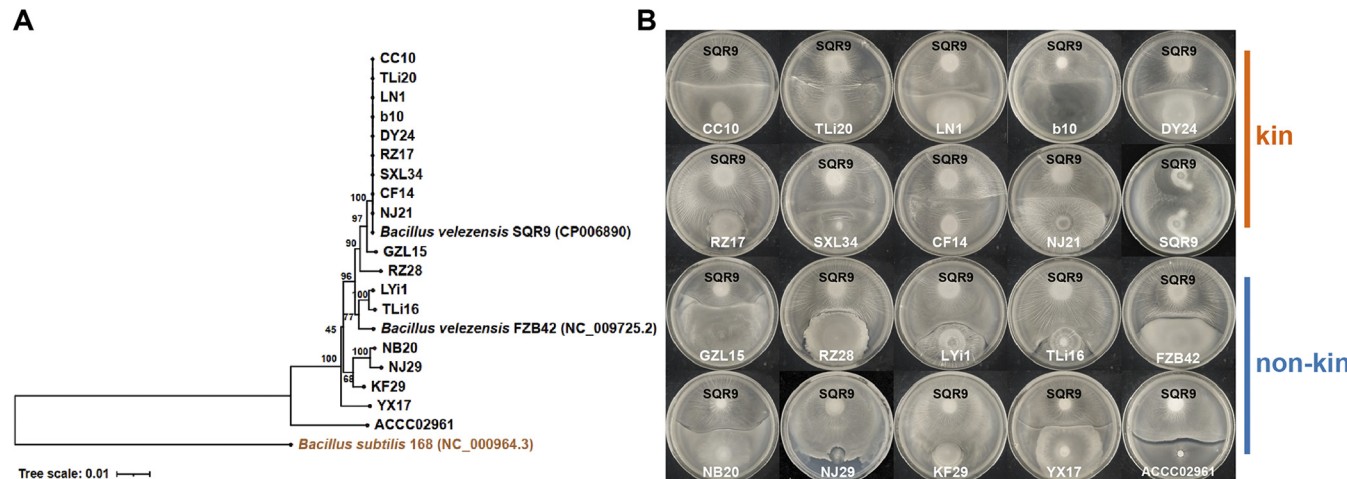

**FIG 2** The recognition phenotype of *B. velezensis* SQR9 and its relative on swarm plate varied from merging to the boundary with their phylogenetic distance. (A) The tree was constructed on *gyrA* gene sequences (2264 bp) using MEGA (v.5.05) for Neighbor-Joining, the *B. subtilis* 168 (NC_000964.3) was selected as the outgroup. And the reliability of clades was tested by the 1000 bootstrap replications. (B) displayed swarm phenotype of *B. velezensis* SQR9 and 20 strains, which were sorted according to the phylogenetic distance between 20 strains and SQR9 on the tree from near to far. The first two rows were the kin of SQR9, which merged with SQR9; the last two rows were the non-kin of SQR9, which formed a boundary with SQR9, and the boundary tends to widen with the increase of the phylogenetic distance between the non-kin strains and SQR9. The results are representative of three experiments.

20 strains (Fig. 2). The phylogenetic tree was constructed basing the housekeeping gene *gyrA* (2264 bp), and 9 strains had the same *gyrA* gene sequences as SQR9 and on a phylogenetic tree branch, the other 10 strains had different *gyrA* gene sequences from SQR9 and were located on different tree branches (Fig. 2A). 9 strains on the same branch with SQR9 formed merge with SQR9 on swarming plates (defined as kin), while 10 strains on different branches without SQR9 formed boundary phenotype with SQR9 (defined as non-kin) (Fig. 2B). Interestingly, among the non-kin strains and SQR9, the boundary width was positively correlated with kinship distance (Fig. S1).

After knowing the kinship of those 20 *B. velezensis* strains, 4 strains with different kinship distances to strain SQR9 were selected: SQR9, FZB42, NB20, and ACCC02961. These 4 strains all possessed complete pericyte flagella and superior swarm motility (Fig. S2A, G-I), and were all merged with themselves on swarming plates (Fig. S3E-H). Four plasmids containing *hag* genes amplified from genome DNA of these 4 strains were transformed into mutant Δ*hag* of SQR9, respectively. The Δ*hag* carrying corresponding plasmids were respectively named Δ*hag-hag*$_{SQR9}$, Δ*hag-hag*$_{FZB42}$, Δ*hag-hag*$_{NB20}$, and Δ*hag-hag*$_{ACCC02961}$. Transmission electron microscope (TEM) images showed that the flagellar filaments were recovered after the transformation of the plasmid, which contains a different *hag* gene (Fig. S2C-F). Interestingly, the Δ*hag-hag*$_{SQR9}$, Δ*hag-hag*$_{FZB42}$, Δ*hag-hag*$_{NB20}$, and Δ*hag-hag*$_{ACCC02961}$ restored a certain motility ability on semi-solid plates, however, the first 2 strains recovered considerably more than the latter 2 on swarm assay medium with 0.5% and 0.7% agar (Fig. 3A to F and Fig. S4). In swarm boundary assay, Δ*hag-hag*$_{SQR9}$ and Δ*hag-hag*$_{FZB42}$ could merge with SQR9wt (Fig. 3G and H and Fig. S3J and K), Δ*hag-hag*$_{NB20}$ was an intermediate phenotype with SQR9wt (Fig. 3I and Fig. S3L), Δ*hag-hag*$_{ACCC02961}$ even formed boundary phenotype with SQR9wt (Fig. 3J and Fig. S3M). In summary, when Δ*hag* mutant of SQR9 was complemented different flagellar filaments from kin and non-kin strains, the swarm phenotypes against wild-type SQR9 are similar to donor strains of *hag* gene.

**Production of bacillunoic acid by Δ*hag* mutant and its flagellin gene complementary strains is similar to SQR9 wild type.** We demonstrated in previous studies that changes in flagella (deletion or replacement with non-kin flagella) resulted in a change in the recognition phenotype (from merge to boundary) of the mutant and complemented strains with SQR9wt on swarm plate. Whether this phenomenon is caused by changes in the flagella or by changes in the secretion of antibiotics needs to be further explored. Our previous work shows that *B. velezensis* SQR9 can secrete a

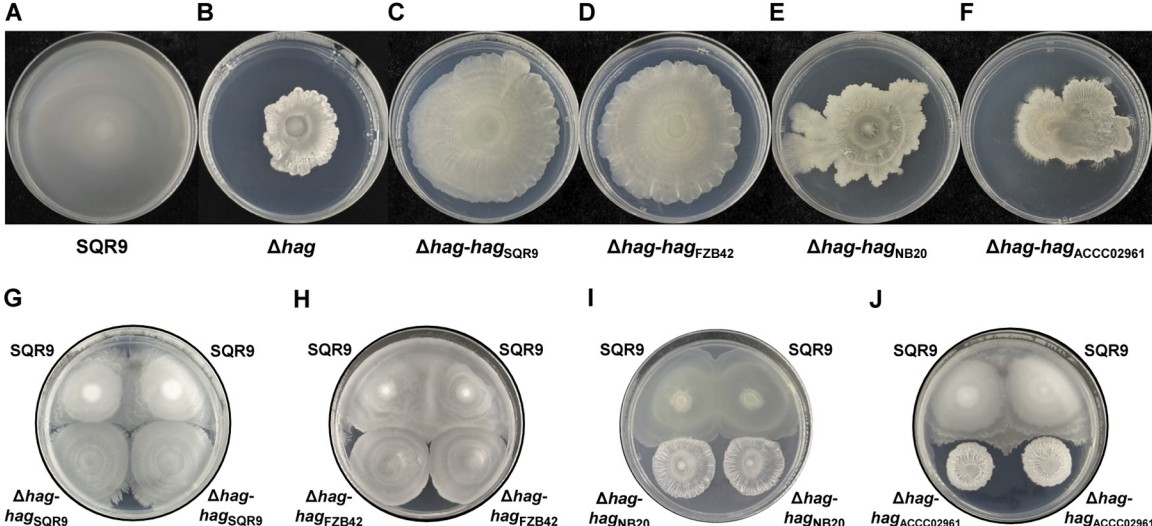

**FIG 3** Wild-type strain SQR9 and its relatives (Δ*hag* mutant of SQR9 assembled with flagellin of kin and non-kin strains) on swarm plates (0.7% gar) show different recognition phenotypes. (A) to (F) The farther the phylogenetic distance between the source strains of flagellin complemented and SQR9, the more difficult it is to restore the swarm ability of flagellin mutant Δ*hag*. The swarm phenotypes of single strain were displayed, including SQR9wt (A), Δ*hag* (flagellin mutant of SQR9) (B), Δ*hag-hag*SQR9 (Δ*hag* carrying *hag* gene of kin strain SQR9) (C), Δ*hag-hag*FZB42 (Δ*hag* carrying *hag* gene of non-kin strain FZB42) (D), Δ*hag-hag*NB20 (Δ*hag* carrying *hag* gene of non-kin strain NB20) (E) and Δ*hag-hag*ACCC02961 (Δ*hag* carrying *hag* gene of non-kin strain ACCC02961) (F). The *hag* genes were complemented into Δ*hag* using plasmids. (G–J) The farther the phylogenetic distance between the source strains of flagellin complemented and SQR9, the more similar the swarming phenotype of SQR9 carrying heterologous flagellin and SQR9wt to that of SQR9wt and Δ*hag*. The swarm phenotypes of two strains were shown, including SQR9wt and Δ*hag-hag*SQR9 (G), SQR9wt and Δ*hag-hag*FZB42 (H), SQR9wt and Δ*hag-hag*NB20 (I), SQR9wt and Δ*hag-hag*ACCC02961 (J). The pictures of the plates were acquired 48 h after inoculation, and the results are representative of three experiments.

variety of antibacterials, including 3 lipopeptides bacillomycin D, fengycin and bacillibactin; three polyketides bacillaene, difficidin, and macrolactin (34, 35). Moreover, a novel antimicrobial fatty acid, named Bacillunoic acids, which showed strong antibacterial against closely related *Bacillus* strains (36). In our preliminary experiments, strains of 6 mutants: Δ*bmy* (synthetic gene mutant of bacillomycin D), Δ*fen* (synthetic gene mutant of fengycin), Δ*dhb* (synthetic gene mutant of bacillibactin), Δ*bae* (synthetic gene mutant of bacillaene), Δ*dfn* (synthetic gene mutant of difficidin), and Δ*mln* (synthetic gene mutant of macrolactin), both merged with SQR9wt (Fig. S5A to F) and showed similar boundary phenotype with non-kin strains ACCC02961 as SQR9wt strain (Fig. S5H to M). Only the ΔGI mutant (the bacillunoic acid synthetic gene mutant) formed boundary phenotype with SQR9wt (Fig. S5G) and formed a weakened boundary phenotype with non-kin strains FZB42, NB20 and ACCC02961 (Fig. 4A to C), indicated that bacillunoic acid is partly involved in the kin discrimination of SQR9.

The bacillunoic acids secreted by SQR9 into the fermentation supernatant can effectively antagonize *B. velezensis* FZB42, making it form an antagonistic circle around the Oxford cup with the fermentation supernatant. We compared the production of bacillunoic acids by using the method of the antagonistic circle diameter measurement (Fig. 4D). Results showed that deletion of flagella or complementation for the various *hag* genes in SQR9 does not affect bacillunoic acid production (Fig. 4E). In addition, we tested the surfactin production of SQR9wt and Δ*hag* mutant in liquid culture using high-performance liquid chromatography. Again, similar production of surfactin was observed between the 2 strains (Fig. S6). These results are consistent with our conclusion that flagellin is partly involved in kin recognition in *B. velezensis* SQR9 without affecting the antibacterial compounds production.

**For *B. velezensis*, the flagellin heterogeneity of non-kin strains is much higher than kin.** The results above suggest that flagellin in *B. velezensis* strains might be involved in the recognition of kin, we next want to investigate whether the structure of flagellin is related to the kinship. We sequenced the *hag* gene sequences of the 20 *B. velezensis* strains

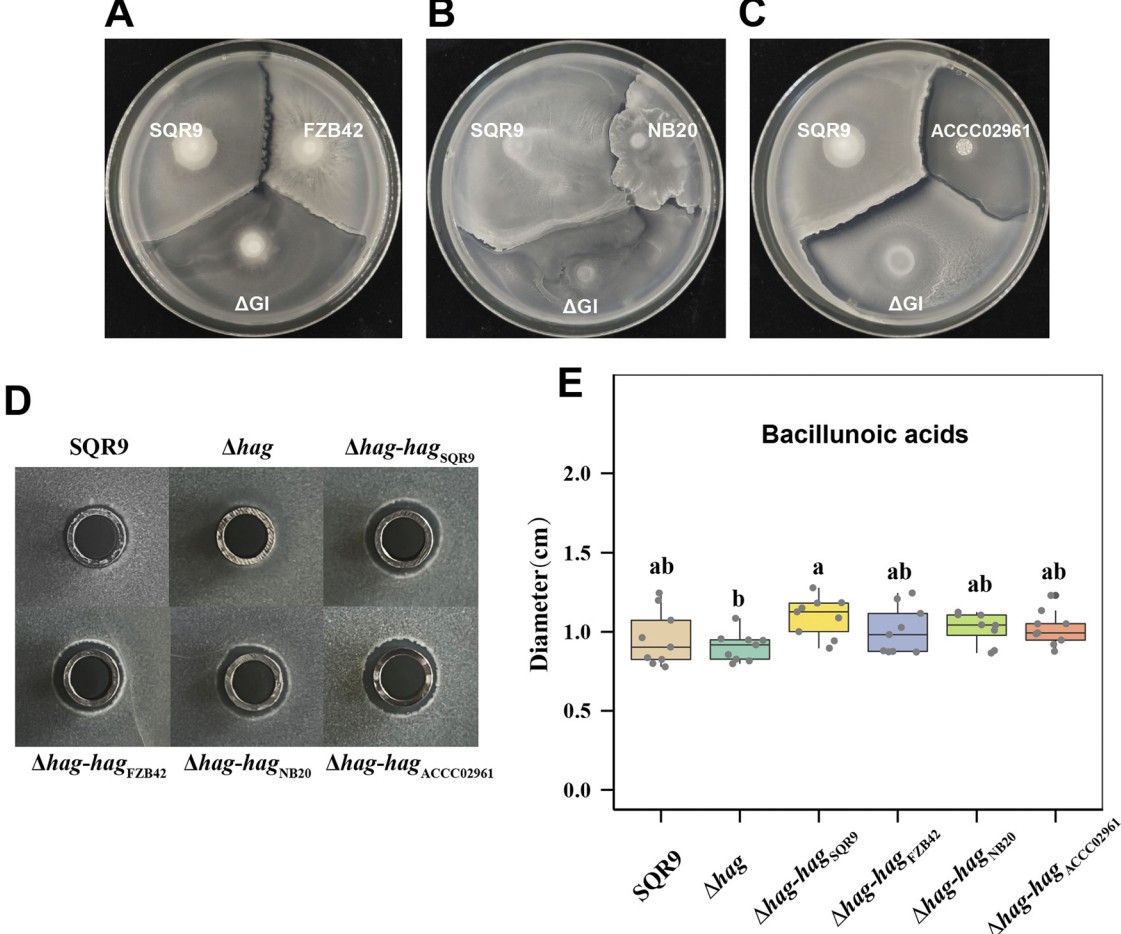

**FIG 4** The deletion and complementation of the *hag* gene of SQR9 had no significant effect on the production of antibacterial substances Bacillunoic acids. Compared with SQR9wt, the ΔGI mutant (the bacillunoic acid synthetic gene mutant) formed a weakened boundary phenotype with non-kin strains FZB42 (A), NB20 (B) and ACCC02961 (C). The ΔGI is a deletion mutant of the gene island in strain SQR9 for the synthesis of bacillunoic acids. The pictures of the plates were acquired 48 h after inoculation, and the results are representative of three experiments. (D) The photo of the antagonism circle assay of *B. velezensis* FZB42 by the fermentation supernatant of strains SQR9wt, Δ*hag*, and four *hag* gene complemented strains (Δ*hag-hag*SQR9, Δ*hag-hag*FZB42, Δ*hag-hag*NB20 and Δ*hag-hag*ACCC02961). (E) The production of Bacillunoic acids in SQR9 wild-type, Δ*hag* mutants, Δ*hag-hag*SQR9, Δ*hag-hag*FZB42, Δ*hag-hag*NB20 and Δ*hag-hag*ACCC02961, was assessed using the antagonism assay of the fermentation supernatant of the tested strain to *B. velezensis* FZB42. The antagonism assays for each strain included nine replicates. The box plots were drawn using R (v.4.0.3), and the analysis of significant differences was performed using Duncan's multiple range tests ($P < 0.05$) on SPSS (v. 25).

and translated them into protein sequences, then analyzed their structural features. Results showed that both ends of the sequences were relatively conservative, and there were only several variant bases, but the middle part of the sequences was very different (Fig. S7). To observe the details of the variable region of the flagellin sequences more clearly, we cut and display the variable region separately (Fig. 5A). The sequences belonging to different strains were sorted according to the *gyrA* gene similarity of the strains and SQR9, that was, the strain ACCC02961 at the bottom had the farthest kinship with SQR9. The strains on a tree branch with SQR9 have the same flagellin sequence as SQR9. Other strains had an extra sequence in the variable region of the flagellin sequence (the length of sequences was 52–57 aa), except for strain FZB42 (Fig. 2A and Fig. 5A). The flagellin sequence variant region of FZB42 was very similar to SQR9, compare to strains on adjacent branches (Fig. 5A). This may be due to individual differences in strains, or the evolutionary rate of the *hag* gene was not strictly consistent with the *gyrA* gene.

To clarify what difference the variable region of the flagellin sequence caused, we selected 4 strains that were known genomes: SQR9, FZB42, NB20, and ACCC02961, then applied the entire *hag* gene sequences to predict the tertiary structure of flagellin and

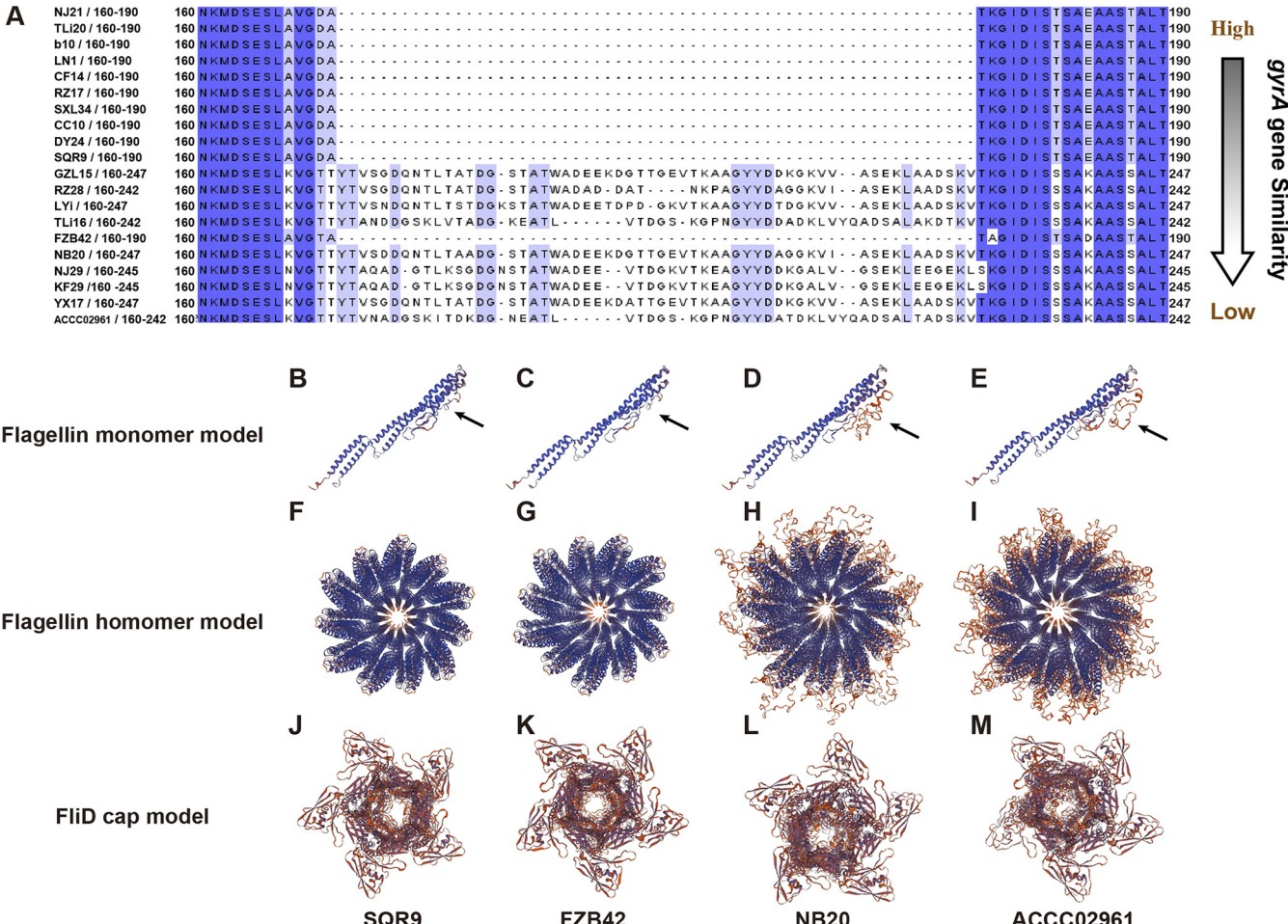

**FIG 5** Flagellin sequence and protein structure varied more between non-kin strains than between kin strains in *B. velezensis*. These pictures showed the variable region of the *flagellin sequences* of 20 *B. velezensis* strains (A) and the protein structure prediction of flagellar monomer (B) to (E), flagellar homomer (F) to (I), and flagellar cap (J) to (M) according to four *B. velezensis* strains: SQR9, FZB42, NB20, and ACCC02961. These prediction structures of flagellin homomer and flagellar cap showed their cross sections from the top view of the filaments. The *hag* gene sequences were aligned using the L-INS-I method of MAFFT (v7.487) and displayed using Jalview (v.2.11.1.5), and the protein structure of flagellin was predicted on the Swiss-Model website.

flagellar filaments by using comparative modeling method based on Swiss-model database. The most similar template matched by the flagellin sequences of four strains was the same template 6t17.1.A, and the details of template matching information were placed in Table S1 (see Table S1 at [https://zenodo.org/record/7131344#.Yzei8thBxPY]). Based on the prediction results of the flagellin monomer, the variable region of the flagellin monomers of the NB20 and ACCC02961, which had more $\beta$-strands and random coils than that of SQR9 and FZB42 (Fig. 5B to E). After the flagellin monomers were assembled into flagellar filaments, the part of $\beta$-strands and random coils was exposed on the periphery of the columnar flagellar filaments (Fig. 5F to I). In addition, we also carried out the protein structure prediction of the flagellar cap structure (Table S1; at [https://zenodo.org/record/7131344#.Yzei8thBxPY]) and found that the flagellar cap structure of the 4 strains was very similar, and they were all composed of 5 protein monomers (Fig. 5J to M). The flagellar cap was first assembled on the flagella, and then helps and regulates the flagellin monomers to gradually assemble into flagellar filaments (37, 38). Therefore, SQR9 can assemble different flagellin monomers into flagellar filaments. Taken together, these results above suggest that flagellin in *B. velezensis* strains might be involved in the recognition of kin.

**The variation of flagellin is higher than other flagellar proteins, and it is not conservative in *B. velezensis*.** In the above analysis, we observed a wider variation in a specific area of the *hag* gene among 20 *B. velezensis* strains, we next want to explore

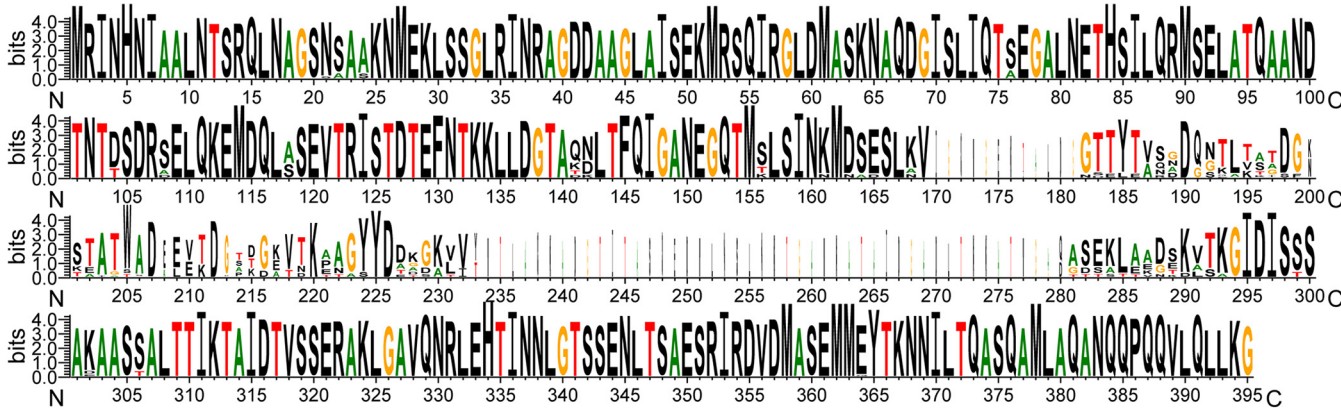

**FIG 6** Flagellin sequences logos exhibit dramatic variation in *B. velezensis*. The variation was mainly concentrated in the 170–293 sites region of flagellin sequences, which contained many large insertions and deletions. These sequences came from 190 *B. velezensis* strains in the NCBI database, their lengths were in the range of 276–384 amino acids, and were aligned using the L-INS-I method of MAFFT (v7.487) and displayed using WebLogo (v.3.7.4).

the flagellin diversity of *B. velezensis*. Therefore, we analyzed the sequence variation of flagellin in 190 *B. velezensis* strains from the NCBI genome database and used several other flagellar structural proteins as a reference.

During alignment, it was found that the middle region of the flagellin sequences (170-293 sites) showed the most variation (Fig. 6), with many large insertions or deletions, roughly divided into 3 lengths: two amino acids, 21 amino acids, and 110 amino acids (Fig. S9; at [https://zenodo.org/record/7086025#.YyR7PaRBxPY]). The divergence of the flagellin sequences was greater than that of the 20 *B. velezensis* strains above (Fig. S7). In contrast with the intermediate variable region of flagellin, both ends of flagellin have fairly conservative sequences, with only a few amino acid sites having variation, and the tail (C-terminal) was more conservative than the head-end (N-terminal) (Fig. 6).

In addition, we also analyzed sequence variations of other flagellar proteins, including extramembrane flagellar structural protein: filament cap protein (encoded by *fliD* gene) (Fig. S8A), junction protein (encoded by *flgK* and *flgL* genes) (Fig. S8B and C), hook cap protein (encoded by *flgD* gene) (Fig. S8D), hook structure protein (encoded by *flgE* gene) (Fig. S8E); intramembrane protein: flagellar rod structure protein (encoded by *flhO* gene) (Fig. S8F). The analysis results showed that the primary sequence homology of these proteins was high (Fig. S8B to F), and only the individual amino acid residues of filament cap protein differed (Fig. S8A). The results above indicate that these structures of the flagella of the intraspecies strains of *B. velezensis* were all extremely conservative, with little variation.

In summary, the specific area of flagellin has large variation within *B. velezensis* strains and the variation degree of flagellin is higher than that of other structural proteins of flagella, whether the N-terminal conserved region or the central variable region, indicating that flagellin evolved faster than other structural proteins of flagella in *B. velezensis*.

## DISCUSSION

In the natural environment, bacteria will use a variety of methods to maximize the benefit of progeny, and kin discrimination is one of them (21). In this study, we investigated whether flagella were involved in kin recognition of *B. velezensis* strains. To clarify the role that flagella play in kin recognition, we performed an exchange experiment of kin and non-kin strain's flagellin on *B. velezensis* SQR9, then detected their swarm phenotype. Our results showed that flagellin heterogeneity (from kin or non-kin strains) affected recognition of *B. velezensis* strains.

The *hag* gene encodes *Bacillus* flagellin, and its absence makes SQR9 unable to synthesize flagellar filaments (39). Electron micrographs showed that mutant Δ*hag* has no

flagella (Fig. 1B). Strains cannot swarm without flagella (Fig. 1E) but can slide, a short-distance migration movement that does not depend on flagella (40), which accounts for the ability of the Δ*hag* community to spread outward from the inoculation site (Fig. 1E and G). When SQR9wt with flagellar filaments encounter mutant Δ*hag* without flagellar filaments, it will detour (Fig. 1G), instead of the same strain as kin to merge (10), which implied that the lack of flagella filaments changed inherent patterns of communication and cooperation between the 2 populations.

The next stage of investigation revolves around 20 strains of *B. velezensis*, and the strains with farther kinship distance from SQR9 had wider swarm boundaries with SQR9 (Fig. 2 and Fig. S1), which was consistent with previous reports (13, 33). Here, we need to note that we used *B. velezensis* as a model given the numerous strains that have been collected. Many plant-growth promoting strains (*Bacillus amyloliquefaciens*, among others) were reclassified as *B. velezensis* (41). Then, flagellin of 4 strains (SQR9, FZB42, NB20, and ACCC02961) with different kinship distances from SQR9 was swapped to SQR9, and the results showed that the swarming phenotype between SQR9wt and SQR9 with heterologous flagellin filaments was related to the flagellin divergence of the 2 strains (Fig. 3G to J). The *hag* mutant strain of SQR9 assembling flagellin from its kin merged with wild type when they encountered each other on swarming plates; however, the *hag* mutant strain of SQR9 assembling flagellin of non-kin strains (NB20 and ACCC02961) forms boundary with wild type. Flagellin region interchange experiments further confirmed that flagella are involved in the kin recognition in *B. velezensis* SQR9.

The execution of outer membrane exchange in *M. xanthus* requires the initial recognition of 2 cell surface proteins, TraA and TraB, and successful outer membrane exchange can only be achieved when they both are present and have the same or similar structures (42, 43). Some strains of $\alpha$-, $\beta$-, and gammaproteobacteria use the CDI (contact-dependent inhibition) system to secrete adhesins for primary recognition with surface receptors of neighboring cells (44, 45). In our study, flagellin played a role in the kin recognition of *B. velezensis* SQR9. However, the same SQR9-body means that they have the same virulence and immune system (11), therefore even if they are identified as non-kin as the difference in flagellin, they cannot attack and kill each other.

Given this, we analyzed the differences between the *hag* gene and flagellin structure between kin and non-kin strains. The results showed that the central region length of *hag* gene sequences of kin and non-kin strains varied greatly, ranging from 52 aa to 57 aa (Fig. 5A), which encodes the $\beta$-sheet and coils on the D2 domains of the flagellin monomer. Research has shown that the D2 domains contribute to the stability of flagellar filaments and that deletion of the domain also affects the primary anti-flagellin responses (46). In the study, the 4 *Bacillus* strains of flagellin did not contain the D3 domain and had 2 types of flagellin: without D2 domain (SQR9 and FZB42), containing D2 domain (NB20 and ACCC02961) (Fig. 5B to I). The lengths of amino acid sequences of flagellin vary widely among bacteria, especially the variable region at the center, ranging from 8 amino acid residues in *Clostridium tetani* to 247 amino acid residues in *Helicobacter pylori* (29). This part of the structure does not participate in the polymerization of flagellin monomers. After the flagellin monomer is assembled into filaments, the structure will be exposed on the periphery of the filaments and directly in contact with the outside environment (29, 47). The wide variation in flagellin structures within species makes them potentially capable of mediating bacterial-bacterial recognition.

Our results showed that flagellin heterogeneity (from kin or non-kin strains) affected recognition of *B. velezensis* strains. (Fig. 5). Such high variability and rapid evolution of flagellin were reported to exist in many species (48, 49), including 2 levels: the first level is the large difference in flagellin variable regions, which may involve gene transfer across phyla; the second level is that there are evolutionary variations in species, similar to the evolution of other conserved genes (49). For *Bacillus*, sequence diversity of flagellin within the same species has been widely investigated. In *Bacillus cereus* and *Bacillus thuringiensis*, flagellin sequence variation is one of the bases for classification (48). Here we found that flagellin sequence variation may plays an important role in kin recognition

**TABLE 1** Microorganisms used in this study

| Strains | Genotype | Reference or source |
|---|---|---|
| *B. velezensis* SQR9 | Wild type | 60 |
| *B. velezensis* FZB42 | Wild type | 61 |
| *B. velezensis* CC10 | Wild type | This study |
| *B. velezensis* TLi20 | Wild type | This study |
| *B. velezensis* LN1 | Wild type | This study |
| *B. velezensis* b10 | Wild type | This study |
| *B. velezensis* DY24 | Wild type | This study |
| *B. velezensis* RZ17 | Wild type | This study |
| *B. velezensis* SXL34 | Wild type | This study |
| *B. velezensis* CF14 | Wild type | This study |
| *B. velezensis* NJ21 | Wild type | This study |
| *B. velezensis* GZL15 | Wild type | This study |
| *B. velezensis* RZ28 | Wild type | This study |
| *B. velezensis* LYi1 | Wild type | This study |
| *B. velezensis* TLi16 | Wild type | This study |
| *B. velezensis* NB20 | Wild type | This study |
| *B. velezensis* NJ29 | Wild type | This study |
| *B. velezensis* KF29 | Wild type | This study |
| *B. velezensis* YX17 | Wild type | This study |
| *B. velezensis* ACCC02961 | Wild type | This study |
| *B. velezensis* SQR9–pUBXC | *B. velezensis* SQR9 with pUBXC, Zeo$^R$ | 62 |
| Δ*hag* | Mutant of *B. velezensis* SQR9, *hag::Em* (Zeo$^R$ Spc$^R$) | This study |
| Δ*bmy* | Mutant of *B. velezensis* SQR9, Δ*bmyD-C* (Zeo$^R$) | |
| Δ*fen* | Mutant of *B. velezensis* SQR9, Δ*fenA-E* (Zeo$^R$) | |
| Δ*dhb* | Mutant of *B. velezensis* SQR9, Δ*dhbA-F* (Zeo$^R$) | |
| Δ*bae* | Mutant of *B. velezensis* SQR9, Δ*baeB-R* (Zeo$^R$) | 63 |
| Δ*dfn* | Mutant of *B. velezensis* SQR9, Δ*dfnA-M* (Zeo$^R$) | |
| Δ*mln* | Mutant of *B. velezensis* SQR9, Δ*mlnA-I* (Zeo$^R$) | |
| Δ*srf* | Mutant of *B. velezensis* SQR9, Δ*srfAA-AD* (Zeo$^R$) | |
| ΔGI | Mutant of bacillunoic acid synthesis gene island *in B. velezensis* SQR9, *GI::Cm* (Zeo$^R$ Cm$^R$) | 36 |
| Δ*hag-hag*$_{SQR9}$ | *B. velezensis* SQR9, *hag::Spc*, pNW33N-*hag*$_{SQR9}$ (Zeo$^R$ Spc$^R$ Cm$^R$) | This study |
| Δ*hag-hag*$_{FZB42}$ | *B. velezensis* SQR9, *hag::Spc*, pNW33N-*hag*$_{FZB42}$ (Zeo$^R$ Spc$^R$ Cm$^R$) | This study |
| Δ*hag-hag*$_{NB20}$ | *B. velezensis* SQR9, *hag::Spc*, pNW33N-*hag*$_{NB20}$ (Zeo$^R$ Spc$^R$ Cm$^R$) | This study |
| Δ*hag-hag*$_{ACCC02961}$ | *B. velezensis* SQR9, *hag::Spc*, pNW33N-*hag*$_{ACCC02961}$ (Zeo$^R$ Spc$^R$ Cm$^R$) | This study |

between *B. velezensis* strains. These indicate that the *hag* gene has a stronger response to environmental pressure. This intraspecific flagellin differential phenotype is likely to be a differential expression of intraspecific species communication.

In conclusion, our research showed that the flagellin contributes to the kin recognition between *B. velezensis* strains. However, the specific identification mechanism remains to be explored. In follow-up experiments, it would be interesting to investigate further whether the communication mechanism is flagellar-flagellar or flagellin-receptor specific in *Bacillus* spp.

## MATERIALS AND METHODS

**Strains information and cultural condition.** A list of strains in the study can be found in Table 1, including 20 strains belonging to *B. velezensis* and several mutants of SQR9 (the strain accession number is 5808 in the China General Microbiology Culture Collection Center, CGMCC, and the genome accession number CP006890 in the National Center for Biotechnology Information [NCBI]). All strains came from Laboratory stock or were isolated from soil (Table 1). All strains were grown at 37°C in low-salt LB (LLB) medium, including 10 g of Tryptone, 5 g of yeast extract, and 3 g of NaCl per L.

**Construction of mutants and flagellin gene complementary strains of *B. velezensis* SQR9.** To delete the *hag* gene in the SQR9 genome, the upstream and downstream regions (1039 bp and 973 bp) that flanked the *hag* gene were amplified from the SQR9 genome, and the primers used to amplify upstream and downstream regions were: up-F (5′-CTCGTCGACATTGACTGCATT-3′) and up-R (5′-CGTTACGTTATTAGTTATGCTAGTGTTAAGAGCCGCGAT-3′), down-F (5′-TATAGCATACATTATACGTGCGCAAG CTAACCAACAGC-3′) and down-R (5′-CGGCATTGGCCGTCAGTTCA-3′). The spectinomycin (Spc) resistance gene was amplified from plasmid P7S6 using the primers spc-F (5′-ATCGCGGCTCTTAACACTAGCATA ACTAATAACGTAACG-3′) and spc-R (5′-GCTGTTGGTTAGCTTGCGCACGTATAATGTATGCTATA-3′), and the amplified regions (1029 bp) contained 18 bp overlap with the upstream and downstream fragments of the *hag* gene, respectively. The mixture volume (50 μL) for amplifying upstream, downstream, and Spc

resistance gene fragments was: 18 $\mu$L of water, 1 $\mu$L of 1 $\times$ Phanta Max Master Mix DNA polymerase (Vazyme), 1 $\mu$L of dNTP mix, 25 $\mu$L of Buffer, 2 $\mu$L of the forward primer, 2 $\mu$L of the reverse primer, and 1 $\mu$L of DNA template. The PCR program was performed under the following conditions: 98°C for 2 min and 32 cycles at 98°C for 10 s, 55°C for 10 s, and 72°C for 4 min.

Upstream, downstream, and *spc* resistance gene fragments were fused using the method of two-step overlapping PCR (50). The mixture volume (25 $\mu$L) for the first step was: 5 $\mu$L of water, 0.5 $\mu$L of 1 $\times$ Phanta Max Master Mix DNA polymerase (Vazyme), 1 $\mu$L of dNTP mix, 12.5 $\mu$L of Buffer, 2 $\mu$L (100 ng) of the upstream fragment, 2 $\mu$L (100 ng) of downstream fragment, and 2 $\mu$L (100 ng) of resistance gene fragment. The PCR program was performed under the following conditions: 98°C for 2 min and 12 cycles at 98°C for 10 s, 50°C for 10 s, and 72°C for 4 min. For the second step, the mixture volume (50 $\mu$L) contained 18 $\mu$L of water, 1 $\mu$L of 1 $\times$ Phanta Max Master Mix DNA polymerase (Vazyme), 1 $\mu$L of dNTP mix, 25 $\mu$L of Buffer, 2 $\mu$L of the primer up-F (the forward primer of the upstream fragment), 2 $\mu$L of the primer up-R (the reverse primer of the downstream fragment), and 1 $\mu$L of product from the first PCR step. In addition, the PCR program was performed under the following conditions: 98°C for 2 min and 32 cycles at 98°C for 10 s, 55°C for 10 s, and 72°C for 4 min.

After purification of the fused fragment of 3 genes (upstream, downstream, and Spc resistance gene), the transformation was conducted by the artificial induction of genetic competence. When SQR9 with plasmid pUBXC (carrying the xylose-inducible comK expression cassette) was cultivated to an $OD_{600}$ of 0.5 in LB medium, 1% (wt/vol) xylose was added. After 1 h of incubation, 20 $\mu$L fused fragment was mixed with 200 $\mu$L SQR9 cells in a 2 mL centrifuge tube and incubated at 37°C for 7 h. Then, cells were plated on LB agar plates including 100 $\mu$g mL$^{-1}$ Spc, and the correct mutants were verified by sequencing (51).

To obtain 4 flagellin gene complemented strains *B. velezensis* SQR9, the *hag* gene of these strains: SQR9, FZB42, NB20, and ACCC02961 were amplified with primers *hag* (F:5′-GCTCTAGAGAAGCGCC TCAGCACGTAGA-3′, R:5′-CGCGGATCCGAGAACCAGGGATCTTTCCGTC-3′) containing 2 cleavage site of restriction endonuclease (XbaI: TCTAGA and BamHI: GGATCC), and the amplification of the *hag* gene fragments was the same as above. Then, the obtained fragments were ligated into the plasmid pNW33N (52) using restriction enzyme ligation technology.

The *hag* gene fragments and plasmid pNW33N (carry chloramphenicol [Cm] resistance gene) were digested with restriction endonucleases: XbaI and BamHI (TaKaRa), and the reaction system and conditions were referred to the instructions on the website of the TaKaRa bio (https://www.takarabiomed .com.cn/Product.aspx?m=20150106133447710028). After that, the *hag* gene fragment with the 2 sticky ends exposed and the plasmid pNW33N was ligated overnight at 16°C, and the enzymatic ligation system was as follows (10 $\mu$L): 5 $\mu$L Solution I (TaKaRa), 1 $\mu$L (100 ng) plasmid pNW33N and 4 $\mu$L (400 ng) *hag* gene fragment. The ligation mixture was transferred into $\Delta hag$ mutant with plasmid pUBXC using the xylose induction method mentioned above. Then, cells were plated on LB agar plates including 100 $\mu$g mL$^{-1}$ Spc and 5 $\mu$g mL$^{-1}$ Cm, and the correct complement strains were verified by extracting plasmid and sequencing. All these mutants and flagellin gene complementary strains are listed in Table 1.

**Swarm assay.** Swarm assays were performed on the 9 cm plates containing B-medium with 0.7% agar at 37°C (13). Strains were grown on solid LLB plates at 37°C for 12 h before use and then transferred to 3 mL of liquid B-medium and shaken overnight at 37°C. The overnight cultures were then diluted to an optical density ($OD_{600}$) of 0.5, and 2 $\mu$L was spotted on the agar plates. The plates with a cover were dried in a laminar flow hood for 30 min, sealed and incubated for 2 days at 37°C, and photographed. Regarding the determination of boundary widths of non-kin strains in swarm boundary assays, each pair of strains included 6 replicates of the swarm boundary phenotype, and each replicate was measured three times using the Image J (v.1.53c) (53). The point plots were drawn using the ggplot2 package and linear regression analysis is performed using the lm() function in R (v.4.0.3).

**Motility test.** To test the motility ability of different strains, 3 kinds of B-medium with 0.3%, 0.5%, and 0.7% agar were selected. The strain preparation and culture conditions were the same as above. Pictures of the plates were acquired 24 h after inoculation. The halo area of the strains was measured in Image J (v.1.53c).

**Determination of surfactin production.** Forty milliliters of sterile supernatant of the tested strains were cultured in Landy medium (54) at 30°C for 60 h, the pH was adjusted to 2.0 with 6 mM HCl, and sat at 4°C overnight. It was then centrifuged to retain the pellet, 4 mL of methanol (LC/MS, Merck) was added to soak for 5 h, and the sample was filtered through a 0.22 $\mu$m membrane to obtain a sterile crude extract, which was stored at 4°C for testing.

Determination of surfactin production of the strains was performed using an HPLC 1200 apparatus (1200 series; Agilent). A high performance liquid chromatography (HPLC) system equipped with an Agilent ZORBAX Eclipse XDB-C18 (250 $\times$ 4.6 mm, 5 $\mu$m) column was operated and maintained at 30°C. A mobile phase mixture consisting of an Acetonitrile and 0.1% (vol/vol) $CH_3COOH$ solution (ratio of 88:12) was pumped in an isocratic mode with a flow rate of 0.84 mL min$^{-1}$. The injection volume of the sample was set at 20 $\mu$L and was detected through a VWD detector at 210 nm. Each analysis was completed within 20 min.

Surfactin standard solution (1000 mg L$^{-1}$) was prepared from 99% pure surfactin (shyuanye). The surfactin substance peaks in the sample to be tested were determined by comparing the chromatographic peak of the sample with the surfactin standard solution, and the yield of surfactin in the sample was characterized by the sum of the absorbance values of the last 3 well-separated surfactin substance peaks. Each sample was tested in triplicate. Analysis of significant differences was performed using independent sample T-Test ($P < 0.05$) on SPSS (v. 25).

**Yield determination of bacillunoic acids.** The bacillunoic acid production of strains SQR9, Δ*hag*, Δ*hag-hag*~SQR9~, Δ*hag-hag*~FZB42~. Δ*hag-hag*~NB20~, and Δ*hag-hag*~ACCC02961~ were evaluated by the inhibitory effect of the fermentation supernatant on the target strain *B. velezensis* FZB42.

Five milliliters of a diluted overnight culture of FZB42 ($\sim$10$^5$ CFU mL$^{-1}$) was spread onto LLB plates (10 $\times$ 10 cm) to be grown as a bacterial lawn. The supernatant of the strain to be tested that was cultured in medium B for 48 h (37°C, 170 rpm), was concentrated 3 times using a centrifugal filter (10 kDa, Amicon Ultra-15), then 180 $\mu$L was added to the Oxford cup on the bacterial lawn, and then the plate was placed at 22°C until a clear zone formed around the Oxford cup. It was then photographed, and the antagonistic circle was measured. The antagonism assays for each strain included 9 replicates. The box plots were drawn using R (v.4.0.3).

**PCR amplification.** Genomic DNA was extracted using omega Bacterial DNA Kit D3350 (Omega, Bio-tek), and the concentration and quality of DNA were assessed using a NanoDrop 2000 spectrophotometer. The *gyrA* and *hag* gene sequences of 20 *B. velezensis* strains were amplified by PCR with primers *gyrA* (F:5′-CAGTCAGGAAATGCGTACGTCCTT-3′, R:5′-GTATCCGTTGTGCGTCAGAGTAAC-3′) and *hag* (F:5′-AGAGTTTGA TCCTGGCTCAG-3′, R:5′-GGTTACCTTGTTACGACTT-3′), and were subsequently Sanger sequenced. The sequences of *gyrA* and flagellin protein of 20 *B. velezensis* strains were in Dataset S1 at [https://zenodo.org/record/7131368#.YzeloNhBxPY] and Dataset S2 at [https://zenodo.org/record/7131381#.YzemJdhBxPY], respectively.

**Phylogenetic analysis.** In this study, the phylogenetic analysis of genes was conducted using MEGA (v.5.05) for Neighbor-Joining (55). The 1000 bootstrap replications tested the clades' reliability. Furthermore, annotation and beautification of trees were achieved through the iTol online site (https://itol.embl.de) (56).

**Electron microscopy.** We cultivated the strains on the LLB solid medium at 37°C for 8 h, placed the plate at an angle, and soaked the fresh colony in sterile deionized water for 2 h, during which we gently shook the plate every 20 min. Then, the strain suspension on the copper net was air dried, the flagella of strains were observed with a HT7700 transmission electron microscope (TEM) that operated at 80 kV, and photographed.

**Protein structure prediction.** The *hag* and *fliD* completed sequences gene of strains SQR9, FZB42, NB20, and ACCC02961 were obtained from the NCBI genome database (SQR9 and FZB42) and sequenced draft genomes (NB20 and ACCC02961). The protein tertiary structure of flagellin monomer, flagellin homomer, and flagellar cap were predicted on the Swiss-Model website (https://swissmodel.expasy.org/).

**Protein sequences analysis of flagellar proteins.** In total, 395 available genomes of *B. velezensis* were downloaded from the NCBI database using the ncbi-genome-download script (https://github.com/kblin/ncbi-genome-download/) (Table S2A, see [https://zenodo.org/record/7131360#.YzeIJ9hBxPY]). The complete genomes were kept, and genomes of the whole genome shotgun were filtered out. Then, the location information of genes (*hag*, *fliD*, *flgK*, *flgL*, *flgD*, *flgE*, and *flhO*) sequences on genomes were obtained by alignment with corresponding genes sequences of SQR9 using NCBI-blast+ (v.2.9.0), and the genes sequences were extracted using Fasta Extract tool in TBtools (v1.0986853) (57). Finally, the 190 genomes of *B. velezensis* that have whole genes were retained and translated into protein sequences using the online tools of the EMBL website (https://www.ebi.ac.uk/Tools/st/emboss_transeq/) for the next analysis (Table S2B, at [https://zenodo.org/record/7131360#.YzeIJ9hBxPY]).

These protein sequences of genes were aligned using the L-INS-I method of MAFFT (v7.487) (https://mafft.cbrc.jp/alignment/software/), and the protein sequences variation of genes were displayed using WebLogo (v.3.7.4) (http://weblogo.threeplusone.com/) (58) and Jalview (v.2.11.1.5) (https://www.jalview.org/) (59).

**Statistics.** Duncan's multiple range tests ($P < 0.05$) of the SPSS version 25.0 (IBM, Chicago, IL, version 25.0) was used for statistical analysis of differences among treatments.

**Data availability.** The accession numbers of the genome sequence of *B. velezensis* SQR9, FZB42, NB20 and ACCC02961 in the NCBI are: CP006890, NC_009725.2, JALJAJ000000000, and JALJAK000000000.

## SUPPLEMENTAL MATERIAL

Supplemental material is available online only.

**FIG S1**, JPG file, 0.4 MB.

**FIG S2**, JPG file, 0.9 MB.

**FIG S3**, JPG file, 1.1 MB.

**FIG S4**, JPG file, 0.4 MB.

**FIG S5**, JPG file, 1.2 MB.

**FIG S6**, JPG file, 0.7 MB.

**FIG S7**, JPG file, 2.8 MB.

**FIG S8**, JPG file, 2.3 MB.

## ACKNOWLEDGMENTS

This work was financially supported by the National Nature Science Foundation of China (31972512, 42090060, and 32072665), the Fundamental Research Funds for the Central Universities (KYXK202009 and KYZZ2022001), and the Central Public-interest Scientific Institution Basal Research Fund (No. Y2022QC15). P.Š. and I.M.-M. were

supported by the Program Grant P4-0116 funded by the Slovenian national research agency (ARRS).

Y.L. and Z.X. designed the study, and Y.L. and R.H. performed the experiments. Y.L., Y.M., Y.C., and P.Š. analyzed the data and created the figures. Y.L., I.M.-M., and Z.X. wrote the first draft of the paper, and P.Š., Z.X., R.Z., Q.S., and I.M.-M. revised the paper.

We declare that we have no conflicts of interest.

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
