## [Reviewer comments · mSystems]

Involvement of flagellin in kin recognition between *Bacillus velezensis* strains

Yan Liu, Rong Huang, Yuqi Chen, Youzhi Miao, Polonca Štefanič, Ines Mandic-Mulec, Ruifu Zhang, Qirong Shen, and Zhihui Xu

Corresponding Author(s): Zhihui Xu, Nanjing Agricultural University

Review Timeline:

Submission Date:	August 17, 2022
Editorial Decision:	August 23, 2022
Revision Received:	September 16, 2022
Accepted:	September 16, 2022

Editor: Matthew Traxler

Reviewer(s): The reviewers have opted to remain anonymous.

Transaction Report:

DOI: <https://doi.org/10.1128/msystems.00778-22>

August 20, 2022

Dr. Zhihui Xu
Nanjing Agricultural University
Nanjing
China

Re: mSystems00778-22 (Flagellin partly involved in kin recognition between *Bacillus velezensis* strains)

Dear Dr. Zhihui Xu:

Thank you for submitting your manuscript to mSystems. We have completed our review and I am pleased to inform you that, in principle, we expect to accept it for publication in mSystems. My request is that you consider changing the title from "Flagellin partly involved in kin recognition between *Bacillus velezensis* strains" to "Involvement of flagellin in kin recognition between *Bacillus velezensis* strains". My suggestion here is based on grammar, and on the fact that I think you have adequately proven flagellin's involvement. If you agree to this change, please make this alteration to the title and resubmit your manuscript.

Preparing Revision Guidelines

Sincerely,

Matthew Traxler

Editor, mSystems

Journals Department
Reviewer comments:

September 16, 2022

Dr. Zhihui Xu
Nanjing Agricultural University
Nanjing
China

Re: mSystems00778-22R1 (Involvement of flagellin in kin recognition between *Bacillus velezensis* strains)

Dear Dr. Zhihui Xu:

Your manuscript has been accepted, and I am forwarding it to the ASM Journals Department for publication. For your reference, ASM Journals' address is given below. Before it can be scheduled for publication, your manuscript will be checked by the mSystems production staff to make sure that all elements meet the technical requirements for publication. They will contact you if anything needs to be revised before copyediting and production can begin. Otherwise, you will be notified when your proofs are ready to be viewed.

Publication Fees:

If you would like to submit a potential Featured Image, please email a file and a short legend to mssystems@asmusa.org. Please note that we can only consider images that (i) the authors created or own and (ii) have not been previously published. By submitting, you agree that the image can be used under the same terms as the published article. File requirements: square dimensions (4" x 4"), 300 dpi resolution, RGB colorspace, TIF file format.

We recognize that the video files can become quite large, and so to avoid quality loss ASM suggests sending the video file via <https://www.wetransfer.com/>. When you have a final version of the video and the still ready to share, please send it to mSystems staff at mssystems@asmusa.org.

Sincerely,

Matthew Traxler
Editor, mSystems

Journals Department
Figure S8: Accept
Figure S2: Accept
Figure S7: Accept
Figure S1: Accept
Figure S6: Accept
Figure S3: Accept
Figure S5: Accept
Figure S4: Accept